# Associations of Leisure-Time Physical Activity Trajectories with Fruit and Vegetable Consumption from Childhood to Adulthood: The Cardiovascular Risk in Young Finns Study

**DOI:** 10.3390/ijerph16224437

**Published:** 2019-11-12

**Authors:** Irinja Lounassalo, Mirja Hirvensalo, Anna Kankaanpää, Asko Tolvanen, Sanna Palomäki, Kasper Salin, Mikael Fogelholm, Xiaolin Yang, Katja Pahkala, Suvi Rovio, Nina Hutri-Kähönen, Olli Raitakari, Tuija H. Tammelin

**Affiliations:** 1Faculty of Sport and Health Sciences, University of Jyväskylä, 40014 Jyväskylä, Finland; mirja.hirvensalo@jyu.fi (M.H.); sanna.h.palomaki@jyu.fi (S.P.); kasper.salin@jyu.fi (K.S.); 2LIKES Research Centre for Physical Activity and Health, 40700 Jyväskylä, Finland; anna.k.kankaanpaa@jyu.fi (A.K.); xiaolin.yang@likes.fi (X.Y.); tuija.tammelin@likes.fi (T.H.T.); 3Methodology Center for Human Sciences, University of Jyväskylä, 40014 Jyväskylä, Finland; asko.j.tolvanen@jyu.fi; 4Department of Food and Nutrition, University of Helsinki, 00014 Helsinki, Finland; mikael.fogelholm@helsinki.fi; 5Research Centre of Applied and Preventive Cardiovascular Medicine, University of Turku, 20014 Turku, Finland, and Department of Clinical Physiology and Nuclear Medicine, Turku University Hospital, 20521 Turku, Finland; katpah@utu.fi (K.P.); suvrov@utu.fi (S.R.); olli.raitakari@utu.fi (O.R.); 6Department of Pediatrics, University of Tampere, 33014 Tampere, Finland, and Tampere University Hospital, 33520 Tampere, Finland; 7Department of Pediatrics, Tampere University and Tampere University Hospital, 33520 Tampere, Finland; nina.hutri-kahonen@uta.fi

**Keywords:** physical activity, diet, trajectory, longitudinal, childhood, adolescence, adulthood

## Abstract

A physically active lifestyle and a diet rich in vegetables and fruits have a central role in promoting health. This study examined the associations between leisure-time physical activity (LTPA) trajectories and fruit and vegetable consumption (FVC) from childhood to middle age. The data were drawn from the Cardiovascular Risk in Young Finns Study with six age cohorts. Participants were 9 to 18 years (*n* = 3536; 51% females) at baseline in 1980 and 33 to 48 years at the last follow-up in 2011. LTPA and FVC were self-reported. LTPA trajectories were identified using latent profile analyses, after which the mean differences in FVC across the trajectories were studied. Active, low-active, decreasingly and increasingly active trajectories were identified for both genders. An additional trajectory describing inactivity was identified for females. Those who were persistently active or increased their LTPA had higher FVC at many ages when compared to their inactive or low-active counterparts (*p* < 0.05). In females prior to age 42 and in males prior to age 24, FVC was higher at many ages in those with decreasing activity than in their inactive or low-active counterparts (*p* < 0.05). The development of LTPA and FVC from childhood to middle age seem to occur in tandem.

## 1. Introduction

In 2010, low fruit intake was ranked the fifth, physical inactivity the tenth, and low vegetable intake the seventeenth global risk factor for disease burden [1]. For example, higher physical activity [2], as well as high fruit, vegetable, and legume intake [3] are associated with a lower risk of cardiovascular diseases and all-cause mortality. Additionally, a physically active lifestyle throughout childhood and adolescence has been found to be a factor preventing obesity in young adulthood [4], while fruit and vegetable intake seems to be inversely, but weakly, associated with weight gain later in life [5]. Since these two behaviors are relevant in regard to health, understanding how they are associated with one another may help in improving public health.

Whereas previous cross-sectional [6] and prospective studies [7] have shown that being physically active is associated with higher consumption of fruits and vegetables, the relationship between the pathways of leisure-time physical activity (LTPA) and fruit and vegetable consumption (FVC) from childhood to adulthood have rarely been researched. Since physical activity [8] and dietary behaviors [9,10] established in childhood and adolescence may track into adulthood, studying the associations between these two behaviors during transition phases from childhood to adolescence and from adolescence to adulthood is important.

A disadvantage of studying the tracking of behaviors is that it does not provide detailed information on subgroups that change their behavior over time. The recent advances in trajectory modelling techniques enable the study of behavioral heterogeneity at different phases of life in a data-driven way [11,12]. For example, in the present instance, it is possible to identify distinctive trajectory classes (i.e., subgroups) of physical activity [13] and FVC [14]. Identifying the key groups of individuals and critical windows during the life course that would be the most receptive to physical activity and dietary promotion would contribute to the enhancement of public health.

The main objective of the present study, with a follow-up lasting over 30 years, was to examine the links between different LTPA trajectories and FVC from childhood to middle age. The study contributes to understanding more profoundly how LTPA develops between and within individuals and how inter- and intra-individual LTPA development is associated with FVC.

## 2. Materials and Methods 

### 2.1. Study Design and Participants

The Cardiovascular Risk in Young Finns Study (YFS) is an ongoing, longitudinal, population-based study whose participants have been randomly selected from five Finnish university cities with medical schools: Helsinki, Kuopio Oulu, Tampere, and Turku. Urban and rural areas in and around these cities were included. The study comprises six age-cohorts born in 1962, 1965, 1968, 1971, 1974 and 1977. At baseline in 1980, 3596 boys and girls aged 3 to18 years participated in the study (83% of those invited, *n* = 4 320). Follow-up studies have been conducted in 1983, 1986, 1989, 1992, 2001, 2007 and 2011. Participation rates in the follow-up studies have been satisfactory, ranging from 2060 to 3596, the main reasons for non-participation being lack of interest in the study, the accompanying person being unable to obtain leave from work, child unwilling to participate and fear of clinical examination [15]. Unexpectedly, many participants lost to follow-up earlier in the study have returned. 

For the present study, the sample was restricted to participants with at least one measurement of both LTPA and FVC. In addition, since the physical activity data on participants under age nine was parent-reported, we restricted the self-reported LTPA data to ages nine to 48 years (*n* = 3536; 51% females). The representativeness of the study population has been studied by comparing the baseline (1980) characteristics between the sample of the year 2001 and those lost to follow-up [15]. The results showed that participants were older and more often females than non-participants. However, no significant differences in physical activity, body mass index (BMI) or parental years of education were observed between participants and non-participants. A detailed description of the YFS and the study protocol has been published earlier [15].

### 2.2. Measurements

*Leisure-time physical activity*. LTPA was assessed eight times between the years 1980 and 2011 (ages 9 to 48) through a self-administered questionnaire. In the years 1980 to 1989, the questionnaire items concerned the frequency and intensity of LTPA, participation in sports-club training, participation in sports competitions, and habitual way of spending leisure-time. In the years 1992 to 2011, the LTPA questionnaire items concerned the frequency and intensity of LTPA, frequency of vigorous LTPA, hours spent on vigorous LTPA, the average duration of an LTPA session, and participation in organized LTPA. All items were first recoded (1 = inactivity or very low activity; 2 = moderately intensive or frequent activity; 3 = frequent or vigorous activity) and then summed to create a physical activity index ranging from 5 to 15 [16,17]. The creation of the original index has been described previously elsewhere [16]. An index value under seven describes inactivity and value of eleven high activity. Six is scored when, for example, participants do not usually experience breathlessness and sweating during physical activity, do not engage in rigorous LTPA, engage in organized physical activity occasionally, and engage in LTPA sessions lasting for 20-40 minutes. Eleven is scored when, for example, participants experience a lot of breathlessness and sweating during physical activity, engage in rigorous LTPA several times and 2-6 hours per week, participate in organized LTPA at least once a week and engage in LTPA sessions lasting for 40-60 minutes.

The criterion validity of the childhood (years 1980-89) and adulthood (year 1992 onward) LTPA index has been tested by studying its correlation with indicators of exercise capacity (hypothetical maximal workload sustainable for 6 minutes) in a subsample (*n* = 102) of YFS participants. The correlations were significant in childhood (girls: *r* = 0.39; boys: *r* = 0.33) and adulthood (women: *r* = 0.49; men: *r* = 0.53) [16]. In addition, the LTPA index correlated significantly with 7-day pedometer data (*r* = 0.24 for total steps; *r* = 0.31 for aerobic steps) [17], and also with accelerometer data (*r* = 0.26-0.45) [18].

*Fruit and vegetable consumption*. FVC was assessed six times using two different self-administered questionnaires. The first questionnaire was used in 1980, 1983, 1986, and 1989 (at ages 9 to 27 years) and consisted of two items: Frequencies of fruit and fruit juice consumption and vegetable consumption separately during the past month. Participants selected one of six response categories: (1) not at all or hardly ever; (2) once or twice a month; (3) once a week; (4) a few times a week; (5) nearly every day; and (6) every day. An FVC index was created by summing the values from the two items. The index ranged from 2 to 12, with high scores indicating high FVC.

In 2007 and 2011 (ages 30 to 48 years), the above questionnaire was replaced with a more comprehensive food frequency questionnaire (FFQ) comprising 131 items on different foods and drinks. The FFQ was developed and validated by the Finnish National Institute for Health and Welfare [19]. Participants reported their monthly, weekly or daily consumption of different food and drink items during the past year. The reported intakes (frequency and portion sizes) of fruits (including fresh and canned fruits and berries) and vegetables (including fresh and canned vegetables, root vegetables, mushrooms, cabbage, pulses and edible bulbs) were first converted into grams per day and then summed to form a variable indicating total daily FVC. Those in the 0.5% of the sample with the highest or lowest scores of the total daily FVC for the years 2007 and 2011 (*n* = 38) were excluded in order to remove the extreme under- and over-reporters from the further analyses.

*Covariates*. Participants’ BMI, total energy intake, education level, and their own and their mothers’ number of years of education were used as covariates. Height was measured with a wall-mounted stadiometer and weight with a digital scale. BMI was calculated as the ratio of weight to the square of height (kg/m^2^). Total energy intake was assessed based on the FFQ in 2007 and 2011 [19]. To exclude under- and over-reporters of total energy intake, all those in the 0.5% of the sample with the highest or lowest scores of total energy intake data for the years 2007 and 2011 (*n* = 38) were excluded from the further analyses. Participants were asked to state education level (primary school, vocational school, high school, and university or equivalent) and their own and their mothers’ number of years of education.

### 2.3. Statistical Analysis

Descriptive statistics were calculated by using IBM SPSS Statistics for Windows, version 24.0 (IBM Corp. Armonk, NY, USA), and expressed as means and standard deviations. Differences in the study variables between males and females were tested by using an independent-samples t-test. Further modelling was performed by using Mplus, version 8.0 [20]. 

Distinct LTPA trajectory classes from childhood to adulthood have been identified from the YFS data in a previous study for males and females separately by using latent profile analysis, which is a type of finite mixture modelling [21]. The modelling was performed again for the present study as the composition of the sample had changed slightly owing to the requirement that each participant had at least one measurement of LTPA and FVC. The statistical modelling of the LTPA trajectories has been described in a previous paper [21] and is presented in Appendix A. All analyses were performed separately for males and females owing to differences in LTPA previously observed between the sexes [21]. Missing data were assumed to be missing at random. Model parameters were estimated by using the full information maximum likelihood method with robust standard errors, thus, enabling the use of all the available data.

After identifying the LTPA trajectories, the mean differences in FVC from age 9 to 48 years across the LTPA trajectory classes were studied utilizing the Bolck-Croon-Hagenaars (BCH) approach [22,23,24]. In the BCH approach, the model estimates for the latent classes (here LTPA) are not affected by the auxiliary variable (FVC), thereby avoiding the class membership changes [24]. First, the BCH weights from the latent profile analysis run with the optimal number of LTPA trajectory classes were saved. BCH weights are group-specific weights computed for each participant during the latent profile model estimation. In the second run, the BCH weights were used as training data, and a multiple group regression model was estimated. FVC was regressed on age-specific covariates within the distinct LTPA trajectory classes, and differences in the regression intercepts (i.e., adjusted means of FVC) across the trajectory classes were studied. When, for example owing to small class size, an error in the computation was reported in Mplus, the residual variances were fixed to a value with the lowest Akaike’s Information Criterion, which indicates the best model fit.

The models were adjusted for the participant’s BMI at all ages, mother’s years of education (at ages 9, 12, 15, 18, and 21), the participant’s education level (at ages 24 and 27), the participant’s years of education (at ages 30, 33, 36, 39, 42, 45, and 48), and total energy intake (at ages 30, 33, 36, 39, 42, 45, and 48). Standardized values of the covariates were used when adjusting the models.

### 2.4. Quality Assessment

To enhance reporting quality, the study was conducted according to the Strengthening the Reporting of Observational Studies in Epidemiology–nutritional epidemiology (STROBE-nut) checklist [25] (Appendix A). The Guidelines for Reporting on Latent Trajectory Studies (GRoLTS) checklist [26] was applied to ensure the quality of the trajectory modelling (Appendix A).

### 2.5. Availability of Data and Materials

The datasets analysed during this study are not publicly available for ethical and legal reasons, but are available from the Publication Committee of the YFS on reasonable request. For more information on dataset access, please contact Professor Olli Raitakari, Project Director of the YFS, University of Turku, Finland, olli.raitakari@utu.fi.

### 2.6. Ethics Approval and Consent to Participate

All subjects gave their informed consent for inclusion before they participated in the study [15]. The study was conducted in accordance with the Declaration of Helsinki, and the protocol was approved by the ethics committees of each of the five participating universities (ETMK:68/1801/2017). 

## 3. Results

### 3.1. Participants and Their Characteristics

The sample size of the present study was 3536 (51% females). All eight LTPA measurements had been completed by 508 participants (14.4%), seven by 579 (16.4%), six by 666 (18.8%), five by 572 (16.2%), four by 453 (12.8%), three by 378 (10.7%), two by 241 (6.8%) and one by 139 (3.9%). For FVC, the corresponding figures were 752 (21.3%), 944 (26.7%), 828 (23.4%), 534 (15.1%), 321 (9.1%), 156 (4.4%) and one (0.03%). Table 1 shows descriptive characteristics and missing data for each study variable at participants’ youngest age, nine years of age, and at 45 years. Since the sample size was small at the participant’s oldest age, 48 years of age, the age of 45 was chosen to describe the descriptive characteristics of the participants at the end of the study.

### 3.2. Fruit and Vegetable Consumption of Males Across the Leisure-Time Physical Activity Trajectories

Four LTPA trajectories were identified for males: Persistently low-active (40.9%), decreasingly active (15.7%), increasingly active (31.1%), and persistently active (12.3%) (Figure 1A). More detailed description of the LTPA trajectories is presented in Appendix A and the selection of the final number of classes is presented in Table 2. The lowest level of FVC was found for males on the persistently low-active trajectory at nearly all ages (Figure 2A,B). Compared to those following the persistently active trajectory, the FVC of the persistently low-active males was significantly lower (*p* = [0.000, 0.015]) at half the ages studied (12, 15, 18, 21, 24, 36, and 39 years) (Table 3). In general, the level of FVC declined during adolescence (age 12-18 years) across all the LTPA trajectories (Figure 2A), but increased among the increasingly active, as well as low active (age 33–42) and decreasingly active males (age 42–48) in adulthood (Figure 2B).

During childhood and adolescence (at ages 9–18), the highest mean values of FVC were found for males on the persistently or decreasingly active trajectory (Figure 2A). However, from age 33 onward, the highest mean values of FVC was observed in the increasingly active males (Figure 2B). When compared to the increasingly active males, those who were persistently active had significantly higher FVC at ages 15 and 18 (*p* = [0.003, 0.032]), after which no significant difference in FVC was observed between these two trajectories (Table 3). The participants on the increasingly active and persistently low-active trajectories showed similar levels of FVC in childhood and adolescence (at ages 9–18). However, the two trajectories showed a significant difference in FVC (*p* = [0.000, 0.020]) in favor of the increasingly active trajectory in adulthood (ages 21, 24, 36, 39, and 45) (Table 3). Concurrently, males on the decreasingly active trajectory showed higher FVC than those on the low-active trajectory up to age 24 (*p* = [0.006, 0.042]), but no longer in middle age (Table 3).

### 3.3. Fruit and Vegetable Consumption of Females Across the Leisure-Time Physical Activity Trajectories

Among females, five LTPA trajectories were identified: Persistently inactive (16.8%), persistently low-active (52.4%), decreasingly active (12.3%), increasingly active (15.1%), and persistently active (3.4%) (Figure 1B). For a more detailed description of the LTPA trajectories and the selection of the final number of classes, see Appendix A and Table 2. The mean values of FVC were the lowest among the females on the persistently inactive trajectory at almost all ages (Figure 3A,B). These values were significantly lower than among those on the persistently active trajectory at ages 9, 15, 21, 39 and 45 (*p* = [0.007, 0.049]) (Table 3).

Figure 3 shows that the FVC was highest among the females on the increasingly active trajectory in middle age (Figure 3B). From age 15 onward, FVC was higher among the females on the increasingly active trajectory when compared to the inactive trajectory (*p* = [0.000, 0.005]) (Table 3). FVC was also higher among the increasingly active than low-active females at ages 15, 21, 24, 27, 42, 45 and 48 (*p* = [0.002, 0.048]). Females following the decreasingly active trajectory had higher FVC than their inactive peers at ages 9, 15, 18, 21, 27, 36, and 39 (*p* = [0.000, 0.041]) or their low-active peers at ages 15, 18, and 27 (*p* = [0.000, 0.044]). However, these differences were no longer evident in late middle age. FVC was higher among females identified to the increasingly active trajectory than among females in the decreasingly active trajectory at ages 33, 42, 45, and 48 (*p* = [0.001, 0.044]), but not in childhood, adolescence or young adulthood.

## 4. Discussion

The aim of this study was to gain insight into the relationship between LTPA trajectories and FVC from childhood to middle age. The findings suggest that FVC is likely to be higher in individuals, especially males, who are persistently active from childhood to adulthood when compared to their persistently, less-active peers. These results mirror the findings from a previous longitudinal study suggesting that physical inactivity and unhealthy diet are predictive of each other in men [27]. While FVC in females under 15 and males under 21 on the increasingly active trajectories resembled that of their inactive and low-active counterparts, the increasers of LTPA ended up having the highest FVC in adulthood. This tendency was particularly evident among females. In turn, FVC was higher among the decreasingly active than inactive females up to age 39 after which their FVC no longer differed from one another. Similarly, after males turned 27, the FVC of the decreasingly active males was no longer significantly higher when compared to their low-active peers. The results are in line with previous findings, both cross-sectional [6] and longitudinal [14,28], confirming that higher FVC and LTPA tend to occur in the same individuals. The results also add to previous knowledge by showing how the changes in FVC closely parallel the distinctive contours of each LTPA trajectory over time at many age points.

The general tendency of FVC was declining across all the LTPA trajectory classes during teenage years after which the tendency changed. The only exception was the females following the decreasingly active trajectory with their FVC declining later. The same declining tendency of FVC during adolescence and young adulthood has been observed in a previous longitudinal study [10]. Even though the tendency of FVC might be declining during teenage years, recent evidence from time trend analyses shows how the overall daily FVC between years 2002–2010 has increased among European and US adolescents aged 11–, 13– and 15 years [29].

The present study suggests that FVC level is associated positively not only with persistent activity level, but also with changes in LTPA from childhood to middle age. For example, an increase in LTPA among the increasingly active males and females was observed in young adulthood simultaneously with the increase in FVC. However, among the increasingly active males, LTPA increased only during young adulthood (ages 18–24) with a momentary high peak of FVC at this age, while another, more continuous change towards higher FVC was not observed until later in adulthood. In Finland, most young adults move from the family home just between 18 and 24 years of age [30]. This sort of transition periods of life may change food choices [31] which might explain the momentary peak of higher FVC among the increasingly active males and possibly also among the persistently active females at the age of 21. In contrast, the increase initiated at the age of 18 in FVC among females in the increasingly active trajectory continued until the end of the study period. Women tend to abandon unhealthy behaviors in young adulthood more often than men [32], which might explain why the favorable co-variance in the two behaviors was continuous for females starting already from the age of 18. Additionally, adult women have more pronounced health beliefs, better nutritional knowledge and attach greater importance to a healthy diet than adult men [33,34,35] which might explain the more consistent increase between the two behaviors among the increasingly active women. These results reflect the findings from a previous longitudinal study concluding that an increase in physical activity, in contrast to a decrease in physical activity, is associated with greater improvements in diet quality [36].

Reciprocally, the FVC of the decreasingly active participants dropped to the level of the persistently inactive participants among females and low-active participants among males in adulthood. Thus, decreasing LTPA may pose an additional health risk, due to simultaneous detrimental changes in diet. All in all, only 22% of women and 14% of men in Finland attain the recommended intake of fruits and vegetables [37] with FVC remaining low also worldwide when compared to the recommended level [38]. However, the decreasingly active males of the current study managed to increase their FVC again after turning 40. Indeed, an increasing trend in FVC was not observed only among the increasingly active participants, but also among the persistently active females and persistently low-active and decreasingly active males in adulthood. An increasing trend in FVC during recent decades has been reported among Finnish adults [39,40]. The improved availability of fruits in Northern Europe [41] and the affordability of fruits and vegetables in high-income countries when compared to low-income countries [38] probably explains this positive trend in FVC. Also, having lunch in a staff canteen has been found to be associated with higher FVC and improved diet among Finnish adult employees [42], suggesting that endeavors put into developing healthier catering services in Finland may also play a role in improving adults’ dietary behavior. Furthermore, the consistent effort put in developing nutrition policy and Finnish nutrition guidelines [43] might have an effect on the favorable development of FVC. For example, when comparing the food pyramid in the national dietary recommendations in 1998 [44] to the food pyramid in the latest recommendations in 2014 [45], the latter recommends higher consumption of fruits and vegetables. Also, the guidelines in 2005 [46] recommended a level of 400 grams of fruits and vegetables per day, while the recommended level is nowadays 500 grams [45]. Diet following Finnish nutrition recommendations has been found to be inversely associated with waist circumference and body fat percentage [47] suggesting that consuming fruits and vegetables according to recommendations improves health.

Previous studies highlight the importance of, e.g., BMI, socioeconomic status, living environment, perceived health, life events, and other factors in explaining levels of physical activity or FVC [38,48,49,50,51,52]. Although the models used in the present study were adjusted for BMI, education and total energy intake (only in adulthood), it should be borne in mind that factors other than the two behaviors studied here may explain why FVC was higher among the persistently and increasingly active than low-active or inactive participants. For example, being on an increasingly or persistently active trajectory may be an indicator of an overall positive health orientation [6], while being on an inactive or low-active trajectory could be an indicator of an overall negative health orientation. A previous study on participants aged 13 to 30 years of age supports this interpretation: The authors identified an overall unhealthy trajectory in which daily fruit intake and regular exercise decreased with higher rates of smoking and inebriation [14]. 

Since the development of the two behaviors seems to occur in tandem, future interventions should study whether to target these two behaviors simultaneously, or is it enough to target one of them and improvements in the other behavior will follow. So far, short term interventions studying the issue have had inconsistent findings with two concluding that improvements in physical activity do not lead to healthier diet [53,54], and another one showing that by increasing physical activity improvements in diet can be achieved, but only among boys [55]. Future studies should also investigate the factors that determine why certain people end up on unfavorable and others on favorable dietary and physical activity trajectories and what are the factors enabling the favorable changes of the two behaviors during the lifespan.

This study has several strengths, including a large sample size comprising six age cohorts, recruitment of participants from across Finland, a 30–year follow-up starting from childhood, and multiple follow-up measurements of LTPA and FVC. Together, these features of the YFS made it possible to study the complex relationship between LTPA and FVC during the life course. The use of a novel, data-driven statistical method enabled the identification of LTPA trajectories in which participants’ LTPA developed similarly, while differing from those in the other trajectories [11]. As recommended in the previous literature, whole foods, here fruits and vegetables [56], were used as units of healthy nutrition [57] instead of specific nutrients (e.g., vitamin C). Moreover, the study utilized the STROBE-nut checklist (Appendix A) and a modified version of the GRoLTS checklist (Appendix A) for ensuring appropriate quality in reporting the results. The GRoLTS checklist was modified as not all the items in the list were suited to latent profile analyses, but instead to latent growth mixture modelling and latent class growth analyses.

This study has its limitations. LTPA and FVC were self-reported, which may produce biased results. There is a possibility for recall bias with light leisure-time physical activities being generally harder to recall than vigorous ones [58]. Social desirability bias, meaning the tendency to over-report desired behaviors and underreport undesired ones, is also present with self-reported data [59]. Self-reported physical activity is commonly estimated higher than objectively measured physical activity [60], and study participants, especially women, tend to overestimate their consumption of foods that are considered to be healthy [61]. After participants turned 30, the method used to assess diet changed from a simple 19–item form to a more comprehensive FFQ. The validity of the latter questionnaire is presumably higher than that of the former one as a previous study showed how a similar comprehensive FFQ as used in the current study performed more accurately than 7– and 16–item questionnaires [62]. However, these biases do not necessarily invalidate the results since the measurement instruments are used for ranking individuals and not analyzing the exact amount of fruits and vegetables consumed nor the exact intensity (low, moderate or vigorous) of physical activity. The sample was representative of the general Finnish population. Hence, the present results cannot be generalized to other populations, especially those in low- or middle-income countries or with diverse ethnic groups.

Trajectory modelling also has its limitations. For example, each trajectory is a description of the subgroup’s mean behavior, and thus, no participant is likely to follow the identified trajectory precisely [63]. This is a reliability issue, especially in the small trajectory classes. Also, the mean FVC across the LTPA trajectories might not be reliable at those five age points where the residual variances of the FVC needed to be fixed to a value with the lowest Akaike’s Information Criterion in order to get a result (see FVC mean values marked with c in Table 3). Especially the persistently active women at the age of 21 and at the age of 48 seemed to have a deviant shift in their FVC mean values (Figure 3). On the other hand, this trajectory class was extremely small (3.4% of the sample) which is why a change in the answers of just a few participants may, indeed, change the mean FVC considerably. Finally, although the final number of LTPA trajectory classes was based on objective index values, subjective interpretation was also used, a procedure that could induce selection bias. 

## 5. Conclusions

A parallel relationship from childhood to middle age was found between the LTPA trajectories and FVC. The results support that of previous longitudinal research showing that LTPA behavior and FVC may facilitate each other [64] and, in turn, decreasing LTPA may be an indicator for an additional health risk, due to simultaneous detrimental changes in diet. While the most recent measurements showed improvements in FVC across many of the LTPA trajectories, a decreasing tendency in FVC was observed across nearly all the LTPA trajectories during teenage years and the inactive, low-active and decreasingly active participants generally showed the lowest levels of FVC in adulthood. Putting effort into adopting or maintaining a physically active lifestyle along with healthy dietary habits, starting from adolescence, could be important for health later in life. To achieve the favorable changes in these behaviors, cross-government and multisectoral approaches that facilitate the integration of physical activity and higher FVC in multiple daily settings are needed.

## Figures and Tables

**Figure 1 ijerph-16-04437-f001:**
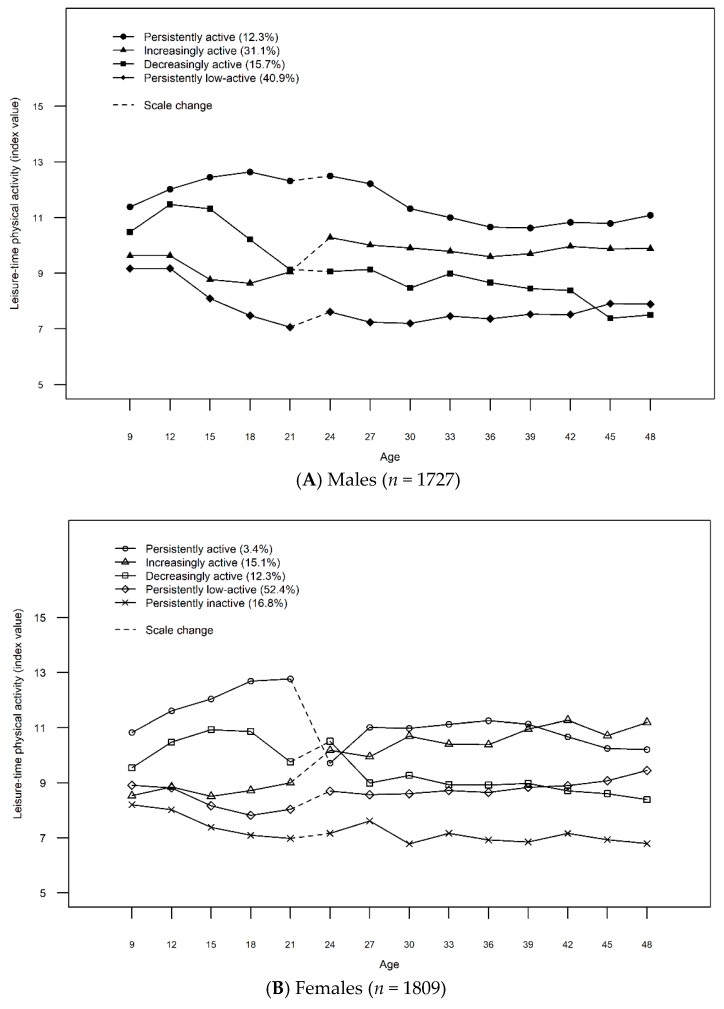
Leisure-time physical activity trajectories for males (*n* = 1727) (**A**) and females (*n* = 1809) (**B**).

**Figure 2 ijerph-16-04437-f002:**
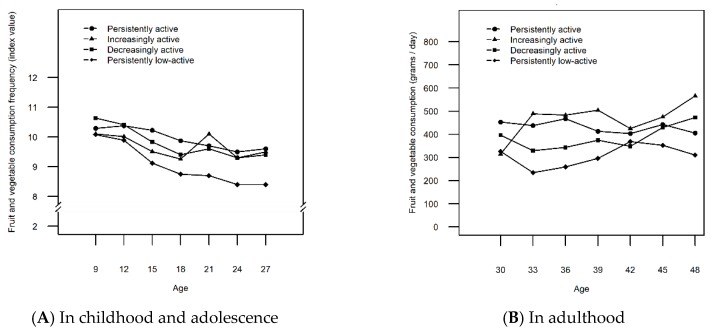
Mean fruit and vegetable consumption in different leisure-time physical activity trajectories among males from ages 9 to 27 years (**A**) and 30 to 48 years (**B**).

**Figure 3 ijerph-16-04437-f003:**
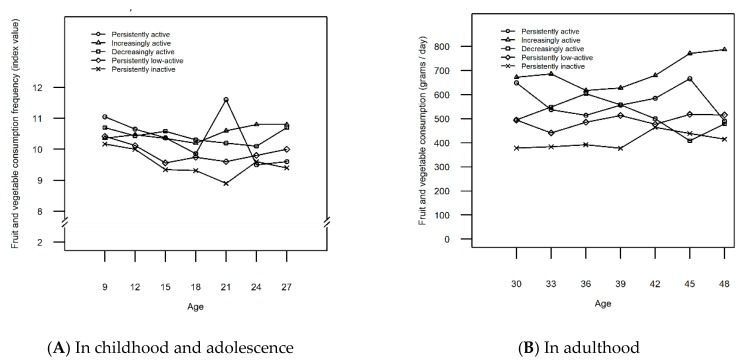
Mean fruit and vegetable consumption in different leisure-time physical activity trajectories among females from ages 9 to 27 years (**A**) and 30 to 48 years (**B**).

**Table 1 ijerph-16-04437-t001:** Descriptive statistics of the study sample at age 9 and 45.

Descriptive Variable at Age 9 and 45	Males		Females		*p* ^a^
	Mean (sd)	*n* (Missing)	Mean (sd)	*n* (Missing)	
**At age 9:**					
LTPA (index, range 5–15)	9.9 (1.6)	798 (100)	8.9 (1.4)	807 (101)	< 0.001
FVC frequency (index, range 2–12)	10.2 (1.7)	778 (120)	10.4 (1.5)	803 (105)	0.010
BMI (kg/m^2^)	16.7 (2.3)	814 (84)	16.7 (2.3)	831 (77)	0.612
Mothers’ education (years)	10.9 (3.3)	798 (100)	10.7 (3.2)	804 (104)	0.315
**At age 45:**					
LTPA (index value, range 5–15)	8.8 (1.9)	305 (239)	8.9 (1.7)	380 (215)	0.405
FVC (grams per day)	390 (206)	297 (247)	494 (214)	367 (228)	< 0.001
Total energy intake (kcal/day)	2629 (799)	297 (247)	2139 (604)	367 (228)	< 0.001
BMI (kg/m^2^)	27.3 (4.2)	318 (226)	26.3 (5.6)	391 (204)	0.004
Education (years)	14.5 (3.6)	315 (229)	15.4 (3.4)	388 (207)	0.001

sd standard deviation; LTPA Leisure-time physical activity; FVC Fruit and vegetable consumption; BMI Body mass index. ^a^ p-value for sex difference (t-test).

**Table 2 ijerph-16-04437-t002:** Latent profile analyses for leisure-time physical activity in males (*n* = 1727) and females (*n* = 1809).

	AIC	BIC	ABIC	VLMR	LMR	BLRT	Entropy	Class Sizes (%) ^a^	AvePP	The Number of Random Start Values and Final Iterations
**Males**										
1	32128	32281	32192	-	-	-	-	-	-	500, 20
2	30662	30897	30760	< 0.001	< 0.001	< 0.001	0.78	73.7%, 26.3%	0.95, 0.90	500, 20
3	30342	30658	30474	0.01	0.011	< 0.001	0.63	43.4%, 40.2%. 16.4%	0.83, 0.78, 0.89	500, 20
**4**	**30139**	**30537**	**30305**	**< 0.001**	**0.001**	**< 0.001**	**0.64**	**40.9%, 31.1%, 15.7%, 12.3%**	**0.80, 0.75, 0.72, 0.85**	**1000, 40**
5	30082	30562	30282	0.198	0.202	< 0.001	0.59	32.2%, 23.6%, 17.2%, 15.7%, 11.3%	0.65, 0.75, 0.72, 0.73, 0.85	1000, 40
6	30021	30583	30256	0.494	0.498	< 0.001	0.58	31.3%, 16.9%, 16.4%, 16.2%, 9.9%, 9.2%	0.64, 0.72, 0.68, 0.69, 0.81, 0.74	1000, 40
**Females**										
1	34757	34911	34822	-	-	-	-	-	-	500, 20
2	33634	33871	33734	< 0.001	< 0.001	< 0.001	0.83	84.0%, 16.0%	0.96, 0.89	500, 20
3	33268	33587	33403	< 0.001	< 0.001	< 0.001	0.64	57.5%, 30.6%, 11.9%	0.82, 0.81, 0.89	500, 20
4	33147	33548	33316	0.615	0.617	< 0.001	0.66	49.8%, 33.5%, 12.5%, 4.2%	0.79, 0.80, 0.77, 0.82	1000, 40
5	33018	33502	33223	0.183	0.184	< 0.001	0.66	52.4%, 16.8%, 15.1%, 12.3%, 3.4%	0.77, 0.79, 0.76, 0.77, 0.87	2000, 80
6	32963	33530	33202	0.322	0.324	< 0.001	0.60	41.7%, 15.7%, 15.0%, 14.1%, 10.0%, 3.5%	0.69, 0.76, 0.67, 0.72, 0.75, 0.85	2000, 80

^a^ Final class proportions for the tested latent class models based on estimated posterior probabilities. AIC Akaike’s information criterion; BIC Bayesian information criterion; BIC sample-size adjusted Bayesian information criterion; VLMR Vuong-Lo-Mendell-Rubin likelihood ratio test; LMR Lo-Mendell-Rubin adjusted LRT test; BLRT Parametric bootstrapped likelihood ratio test. AvePP Average posterior probabilities for most likely latent class membership. The class solution considered optimal is presented in bold.

**Table 3 ijerph-16-04437-t003:** Mean fruit and vegetable consumption across the leisure-time physical activity trajectories.

	Mean FVC in LTPA classes in 1980–1989 ^a^ (SE)							Mean FVC in LTPA classes in 2007–2011 ^b^ (SE)						
**Age in years**	9	12	15	18	21	24	27	30	33	36	39	42	45	48
Sample size, males	777	1022	957	837	487	491	325	140	244	240	287	276	296	116
Sample size, females	802	1038	1068	987	614	618	443	156	323	341	362	375	365	169
**Males:**														
1 Persistently active	10.3 (0.2)	10.4 (0.2)	10.2 (0.2)	9.9 (0.2)	9.7 (0.3)	9.5 (0.3)	9.6 (0.6)	452.8 (55.7)	437.9 (92.6)	467.5 (37.5)	413.1 (35.0)	403.1 (55.8)	442.7 (67.6)	405.3 (101.4)
2 Increasingly active	10.1 (0.1)	10.0 (0.2)	9.5 (0.2)	9.3 (0.2)	10.1 (0.3)	9.3 (0.3)	9.5 (0.6)	315.1 (48.9)	488.7 (87.3)	483.1 (34.3)	504.1 (47.8)	424.8 (33.1)	475.1 (33.5)	565.8 (109.3)
3 Decreasingly active	10.6 (0.2)	10.4 (0.2)	9.8 (0.2)	9.4 (0.3)	9.6 (0.3)	9.3 (0.3)	9.4 (0.8)	396.4 (59.2)	329.2 (68.9)	343.3 (54.3)	374.4 (47.4)	347.9 (74.0)	430.0 (75.7)	472.5 (99.5)
4 Persistently low-active	10.1 (0.1)	9.9 (0.1)	9.1 (0.1)	8.7 (0.1)	8.7 (0.2)	8.4 (0.2)	8.4 (0.3)	326.1 (57.9)	234.4 (115.5)	259.3 (41.4) ^c^	295.8 (22.5)	368.6 (23.4)	352.4 (22.0)	310.2 (61.3)
Statistically significant mean differences in FVC between LTPA classes 1–4 (Z-score test)			2 < 1 **	2 < 1 *	4 < 3 *	4 < 3 *				3 < 2 *				
		4 < 3 *	4 < 3 **	4 < 3 *	4 < 2 ***	4 < 2 **				4 < 2 ***	4 < 2 **			
		4 < 1 *	4 < 1 ***	4 < 1 ***	4 < 1 **	4 < 1 **				4 < 1 **	4 < 1 **		4 < 2 **	
**Females:**														
1 Persistently active	11.0 (0.3)	10.7 (0.3)	10.4 (0.3)	9.9 (0.4)	11.6 ^c^ (0.8)	9.5 (0.6)	9.6 (0.7)	649.2 (156.5)	537.2 (96.5)	514.2 (46.4)	556.3 (83.1)	584.8 (103.8)	666.5 (75.7)	490.4 ^c^ (58.0)
2 Increasingly active	10.4 (0.2)	10.5 (0.2)	10.4 (0.2)	10.2 (0.3)	10.6 (0.4)	10.8 (0.3)	10.8 ^c^ (0.3)	672.8 (90.6)	686.4 (67.2)	617.7 (52.5)	628.2 (67.1)	680.1 (59.1)	771.1 (73.0)	787.2 ^c^ (75.0)
3 Decreasingly active	10.7 (0.2)	10.4 (0.1)	10.6 (0.2)	10.3 (0.2)	10.2 (0.5)	10.1 (0.3)	10.7 (0.3)	495.5 (69.7)	548.8 (55.7)	603.7 (43.6)	557.7 (53.3)	499.5 (65.5)	408.3 (82.3)	478.3 (111.0)
4 Persistently low-active	10.4 (0.1)	10.1 (0.1)	9.6 (0.1)	9.7 (0.1)	9.6 (0.2)	9.8 (0.1)	10.0 (0.2)	494.4 (47.0)	441.3 (68.5)	485.3 (37.4)	513.5 (37.2)	476.8 (30.1)	517.8 (24.4)	515.4 (39.9)
5 Persistently inactive	10.2 (0.2)	10.0 (0.2)	9.3 (0.2)	9.3 (0.2)	8.9 (0.3)	9.6 (0.3)	9.4 (0.4)	378.1 (233.4)	383.3 (73.8)	392.3 (50.7)	377.4 (32.1)	464.4 (41.4)	438.8 (40.7)	415.4 (74.1)
Statistically significant mean differences in FVC between LTPA classes 1–5 (Z-score test)			4 < 1 *											
			4 < 2 **		4 < 1 *								3 < 1 *	
			4 < 3 ***		4 < 2 *		4 < 2 *				5 < 1 *		3 < 2 **	1 < 2 **
			5 < 1 **	4 < 3 *	5 < 1 *		4 < 3 *				5 < 4 *	3 < 2 *	4 < 2 **	3 < 2 *
	5 < 1 *		5 < 2 ***	5 < 2 **	5 < 2 ***	4 < 2 **	5 < 2 **		3 < 2 *	5 < 2 **	5 < 2 **	4 < 2 **	5 < 1 **	4 < 2 **
	5 < 3 *		5 < 3 ***	5 < 3 **	5 < 3 *	5 < 2 **	5 < 3 **		5 < 2 **	5 < 3 **	5 < 3 **	5 < 2 **	5 < 2 ***	5 < 2 **

FVC Fruit and vegetable consumption; LTPA Leisure-time physical activity. * *p* < 0.05; ** *p* < 0.01; *** *p* < 0.001. ^a^ At ages 9-27 FVC was expressed with index (range: 2-12). The model was adjusted for BMI at ages 9-27, for mothers’ education (years) at ages 9-18, and for participants’ education (level) at ages 21-27. ^b^ At ages 30-48 FVC was expressed in grams of fruits and vegetables consumed in a day. The model was adjusted for participants’ education (years), BMI, and total energy intake at ages 30-48. ^c^ The residual variances were fixed to a value with lowest Akaike’s Information Criterion.

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
