# Peer review of "Associations of Leisure-Time Physical Activity Trajectories with Fruit and Vegetable Consumption from Childhood to Adulthood: The Cardiovascular Risk in Young Finns Study"

_ijerph, 2019, doi:10.3390/ijerph16224437_

Round 1

Reviewer 1 Report

Thank you for the opportunity to review this well written paper which addresses two important health-related behaviours. The results contribute to the understanding of the development of physical activity and dietary behaviours in important periods of life.

Line 44 ff  

It would be helpful if two aspects would be described in more detail at the beginning of the article: 1.) Why it is important to study the association between PA and diet (e.g. both behaviours are relevant in regard to the development of obesity); 2.) Why is the transition from adolescent to adulthood a critical and important period in life to study this association?

Line 62-63                    

“With previous findings […] and alcohol consumption [13]” I would suggest to delete this part of the sentence because it does not contribute to understand the study aims.

Line 82                          

Please explain what is meant with “full participants”.

Line 247

Does “At the end of the study period” mean “in older participants”? If yes, it would be easier to understand using “older participant”.

Line 258

The word “lifelong” does not fit when the age range is from 9 to 48 years

Line 265-267  

This is quite a long and complex sentence. Please simplify it. Does “less active peers” refers to the persistently low-active and inactive groups? Where does “after which” refers to?

Line 272-282 

I was surprised to read about these results in the discussion even though they are not described in the results section (only in the supplement file). I suggest removing this part, especially since the discussion section is quite long already.

Line 283-295               

Same as described before: These results are not described before and do not essentially contribute to the aim of the paper. I would suggest deleting this section.

Line 301                        

“recent evidence from time trend analyses shows…” would be more precise

Line 316-324               

The arguments in line 319-324 do not explain why this peak only occurs in subgroups of LTPA, which are even different ones between males and females. At least in women, the peak looks more like an outlier which should not be over interpreted because of the small sample size of this group.

Line 379 ff                    

What are the expected consequences/limitation of self-reported data? E.g. unprecise data due to recall bias and social desirability bias. I suggest combining the first and third limitation described.

Line 379-382               

Please describe how the second limitation may affect the results.

Line 386                        

I would suggest to delete “in epidemiology perspective” and replace “where” with “when”.

Line 405 ff                    

At the present, it is unknown which behaviours influences the other one or if a third component (e.g. positive health orientation like stated in line 353) is responsible for the paralleled development of LTPA and FVC. So I would suggest to reword the last part of the sentence: “…,in turn, decreasing LTPA may be an indicator for an additional health risk…”

Line 411 ff                    

The recommendation for PA promotion is quite general and lacking in content. In addition, central government might not solve this problem by itself. It needs a cross-government and multisectoral approach like stated in the Global Action Plan on Physical Activity 2018-2030 by the WHO. A more precise recommendation like “facilitate the integration of PA into multiple daily settings” would be a better end of this article.

Author Response

First of all, I would like to thank you for your valuable and very useful comments that helped us to improve our manuscript and make it more concise. Than kyou also for the time you took for reviewing our manuscript. I have written our replies to your comments below with italic letters and marked the changes by underlining them in yellow in the manuscript.

Thank you for the opportunity to review this well written paper which addresses two important health-related behaviours. The results contribute to the understanding of the development of physical activity and dietary behaviours in important periods of life.

Line 44 ff  

It would be helpful if two aspects would be described in more detail at the beginning of the article: 1.) Why it is important to study the association between PA and diet (e.g. both behaviours are relevant in regard to the development of obesity); 2.) Why is the transition from adolescent to adulthood a critical and important period in life to study this association?

New references and more detailed justification for the study has been added in the intro (see paragraphs 1, 2 and 3).

Line 62-63                    

“With previous findings […] and alcohol consumption [13]” I would suggest to delete this part of the sentence because it does not contribute to understand the study aims.

It has been deleted.

Line 82                          

Please explain what is meant with “full participants”.

I apologize for the unclear term. The sentence has been rephrased as follows: “The representativeness of the study population has been studied by comparing the baseline (1980) characteristics between the sample of the year 2001 and those lost to follow-up.”

Line 247

Does “At the end of the study period” mean “in older participants”? If yes, it would be easier to understand using “older participant”.

“At the end of the study period” has been changed to “middle age”.

Line 258

The word “lifelong” does not fit when the age range is from 9 to 48 years.

“Lifelong” has been changed to “from childhood to middle age”.

Line 265-267  

This is quite a long and complex sentence. Please simplify it. Does “less active peers” refers to the persistently low-active and inactive groups? Where does “after which” refers to?

The complex sentence has been divided into two sentences: “In turn, FVC was higher among the decreasingly active than inactive females up to age 39 after which their FVC no longer differed from one another. Similarly, after males turned 27, the FVC of the decreasingly active males was no longer significantly higher when compared to their low-active peers.”

Line 272-282 

I was surprised to read about these results in the discussion even though they are not described in the results section (only in the supplement file). I suggest removing this part, especially since the discussion section is quite long already.

Good suggestion. Thank you. I have deleted this paragraph.

Line 283-295               

Same as described before: These results are not described before and do not essentially contribute to the aim of the paper. I would suggest deleting this section.

I have deleted this paragraph as well. I only relocated one sentence from that paragraph (“Generally women have more pronounced health beliefs, better nutritional knowledge and attach greater importance to a healthy diet than men”) to the third paragraph of the Discussion.

Line 301                        

“recent evidence from time trend analyses shows…” would be more precise

Good point. This has been changed as suggested.

Line 316-324               

The arguments in line 319-324 do not explain why this peak only occurs in subgroups of LTPA, which are even different ones between males and females. At least in women, the peak looks more like an outlier which should not be over interpreted because of the small sample size of this group.

This paragraph has been deleted and the issue has been discussed in limitations. I only kept one part of the paragraph (“Transition periods of life, for example, moving from family home, may change food choices.”) and placed it to the third paragraph of the Discussion.

Line 379 ff                

What are the expected consequences/limitation of self-reported data? E.g. unprecise data due to recall bias and social desirability bias. I suggest combining the first and third limitation described.

They have been combined and new references have been added concerning the bias related to self-reported data. The limitations has been rewritten as follows: “LTPA and FVC were self-reported which may produce biased results. There is a possibility for recall bias with light leisure-time physical activities being generally harder to recall than vigorous ones [58]. Social desirability bias, meaning the tendency to over-report desired behaviours and underreport undesired ones, is also present with self-reported data [59]. Self-reported physical activity is commonly estimated higher than objectively measured physical activity [60], and study participants, especially women, tend to overestimate their consumption of foods that are considered to be healthy [61].”

Line 379-382               

Please describe how the second limitation may affect the results.

The comparison of the validity concerning the first and second diet questionnaire has been described as follows: “After participants turned 30, the method used to assess diet changed from a simple 19-item form to a more comprehensive FFQ. The validity of the latter questionnaire is presumably higher than that of the former one as a previous study showed how a similar comprehensive FFQ as used in the current study performed more accurately than 7- and 16-item questionnaires [62]. However, these biases do not necessarily invalidate the results since the measurement instruments are used for ranking individuals and not analysing the exact amount of fruits and vegetables consumed nor the exact intensity (low, moderate or vigorous) of physical activity.”

Line 386                        

I would suggest to delete “in epidemiology perspective” and replace “where” with “when”.

“In epidemiology perspective” has been replaced with “when”.

Line 405 ff                    

At the present, it is unknown which behaviours influences the other one or if a third component (e.g. positive health orientation like stated in line 353) is responsible for the paralleled development of LTPA and FVC. So I would suggest to reword the last part of the sentence: “…,in turn, decreasing LTPA may be an indicator for an additional health risk…”

Excellent idea. The sentence has been changed accordingly.

Line 411 ff                    

The recommendation for PA promotion is quite general and lacking in content. In addition, central government might not solve this problem by itself. It needs a cross-government and multisectoral approach like stated in the Global Action Plan on Physical Activity 2018-2030 by the WHO. A more precise recommendation like “facilitate the integration of PA into multiple daily settings” would be a better end of this article.

The last sentence has been changed as follows: “To achieve the favourable changes in these behaviours, cross-government and multisectoral approaches that facilitate the integration of physical activity and higher FVC in multiple daily settings are needed.”

Reviewer 2 Report

The authors did an excellent job presenting the research.  The introduction is revealing and keeps the interest leading to the study purpose. The research design and method were explained explicitly and in detail. Statistics seem to be robust and the results were well organized and illustrated with appropriate tables and graphs.

Discussions are demonstrated with good reasoning. The findings and explanations are supported with existing research evidence. The authors were able to provide analytical and logical arguments to the current findings. The strengths and limitations are to the point. Overall, it’s a pleasure to read. 

Author Response

I am glad and flattered for receiving this feedback. It is a pleasure to get your positive notions of the hard work we have done for this manuscript. Thank you for your comments.

Reviewer 3 Report

I enjoyed reading your manuscript. I am sure all of this took lots of time.

My one main suggestion unless I missed it in the manuscript is telling the reader over time how many participants were meeting the WHO - physical activity recommendations as children/youth and then in adulthood.

It seems a value of say 10 on the LTPA index would mean one thing when under the age of 18 and another when 18+.

The figures should help us understand what the values mean by <18 and 18+ in terms of meeting the WHO - physical activity recommendations.

Small items are spacing issues for instance line 115; 178; and 3 536 line 72 and a few other places - why not just 3536?

Author Response

First of all, I would like to thank you for your comments and the time you took for reviewing our manuscript. I am glad you found our manuscript to be interesting. I have written my replies to your comments below with italic letters.

My one main suggestion unless I missed it in the manuscript is telling the reader over time how many participants were meeting the WHO - physical activity recommendations as children/youth and then in adulthood. It seems a value of say 10 on the LTPA index would mean one thing when under the age of 18 and another when 18+. The figures should help us understand what the values mean by <18 and 18+ in terms of meeting the WHO - physical activity recommendations.

Thank you for your observation. When this prospective study was initiated in the beginning of the 1980s, there were not yet such PA recommendations as we have now. Most of the original LTPA questions have been kept as they are in order to compare the answers between measurement years. Since LTPA behavior changes slightly when reaching adulthood only some minor changes have been done in the questionnaire when the participants reached adulthood. This is why it is rather difficult to try and analyze the self-reported answers in relation to the current PA recommendations. The self-reported LTPA data from the Young Finns Study is not ideal for analyzing whether a person is meeting the PA recommendations or not, but it is useful for ranking individuals within the study in relation to one another.

We do not address this issue in the manuscript in such detail as here, but we do describe the LTPA index (see Measurements, paragraph one and the references in the paragraph), talk about the validation of the index (see Measurements, paragraph two), and we have strengthened the discussion about the limitations of self-reported PA (see limitations in the Discussion). I hope this answer is satisfactory to you.

Small items are spacing issues for instance line 115; 178; and 3 536 line 72 and a few other places - why not just 3536?

The spacing has been changed accordingly.